# Progesterone Receptor Signaling in the Uterus Is Essential for Pregnancy Success

**DOI:** 10.3390/cells11091474

**Published:** 2022-04-27

**Authors:** Dominique I. Cope, Diana Monsivais

**Affiliations:** 1Department of Pathology & Immunology, Baylor College of Medicine, Houston, TX 77030, USA; 2Center for Drug Discovery, Baylor College of Medicine, Houston, TX 77030, USA

**Keywords:** progesterone, progesterone receptor, endometrium, preterm labor, menstruation, decidualization

## Abstract

The uterus plays an essential role in the reproductive health of women and controls critical processes such as embryo implantation, placental development, parturition, and menstruation. Progesterone receptor (PR) regulates key aspects of the reproductive function of several mammalian species by directing the transcriptional program in response to progesterone (P4). P4/PR signaling controls endometrial receptivity and decidualization during early pregnancy and is critical for the establishment and outcome of a successful pregnancy. PR is also essential throughout gestation and during labor, and it exerts critical roles in the myometrium, mainly by the specialized function of its two isoforms, progesterone receptor A (PR-A) and progesterone receptor B (PR-B), which display distinct and separate roles as regulators of transcription. This review summarizes recent studies related to the roles of PR function in the decidua and myometrial tissues. We discuss how PR acquired key features in placental mammals that resulted in a highly specialized and dynamic role in the decidua. We also summarize recent literature that evaluates the myometrial PR-A/PR-B ratio at parturition and discuss the efficacy of current treatment options for preterm birth.

## 1. Evolutionary Dynamics Shaped the Decidual Response to Progesterone

### 1.1. Decidualization Is a Critical Process in Placental Mammals

Therian mammals, which include marsupials and eutherians (“placental mammals”), evolved the ability to give rise to offspring via live birth [1]. Both marsupials and eutherians develop placentas; however, those from eutherian mammals tend to be more complex and invasive and include erosion of the maternal luminal epithelium by trophoblast [2]. Decidualization is the differentiation of endometrial stromal cells in response to a biological signal originating from the mother or fetus [3,4]. This cellular specialization/differentiation is a feature shared among certain eutherian mammals that experience invasive placentation, due in part to the trophoblast’s direct contact with the endometrial stroma. It is becoming clear that a decidual reaction is critical for suppressing the inflammatory reaction that is typically engaged at the time of implantation and that it is required for the development of a healthy placenta [5].

Extensive transcriptional changes in decidualizing cells drive the differentiation of fibroblast-like endometrial stromal cells into functional decidual cells that can then recruit immune cells, remodel the vasculature, and stimulate the glandular system of the endometrium [5]. The importance of this process to successful pregnancy outcomes is underscored by the fact that defects in decidualization underlie many pregnancy-related problems, such as recurrent pregnancy loss and pre-eclampsia [6,7]. The following sections describe the evolutionary changes that have shaped progesterone receptor (PR) action in the decidua, the presence and consequences of PR variants in the genome, and how these have evolved to control specific processes during early pregnancy. Further sections discuss the basic properties of PR action by focusing on mouse models, genome-wide binding studies, and the role of PR action/withdrawal during labor and menstruation.

### 1.2. Evolution of Progesterone Receptor Action in the Decidua

Recent studies have outlined the evolutionary history of the decidual transcriptome [8,9,10,11]. These studies have shown that ancient transposable elements are highly enriched in the cis-regulatory regions of genes that drive decidualization [8,10]. These ancient transposable elements have conferred progesterone response to decidualizing stromal cells by remodeling the PR binding architecture across the genome [8]. Specifically, it was shown that ancient transposable elements are enriched in the PR binding sites of genes that increase strongly during endometrial stromal cell decidualization. The PR binding sequences of master regulators of decidualization, such as COUP-TF/NR2F1, and members of the FOX, HOX, and GATA gene families were also found to be enriched with sequences of ancient transposable elements, supporting the finding that ancient transposable elements gave rise to regulatory DNA regions that are critical for decidualization. Therefore, transposable elements created novel P4-sensitive transcription factor binding sites that increased the decidual cells’ sensitivity to progesterone and contributed to the evolution of decidualization [8].

Similar evolutionary dynamics have been demonstrated for the heart and neural crest derivatives 2 (*HAND2*) gene, another master regulator of decidualization [9,12]. At the molecular level, the ability of the stromal cell to decidualize arose from its ability to differentiate in response to ovarian progestins that then activate the cyclic AMP/protein kinase A pathway. By taking an evolutionary biology approach and using the opossum as an outgroup (i.e., a placental mammal that is incapable of decidualization), it was identified that PGE2-dependent activation of its receptor PTGER2 is the signaling pathway underlying the origins of decidualization [11]. Therefore, a series of key evolutionary events, partially driven by transposable elements, conferred placental mammals with a unique sensitivity to progesterone during decidualization. This sensitivity to progesterone is driven by the transcriptional activity of PR and has shaped the fetal–maternal interface during early pregnancy in mammals (Figure 1).

### 1.3. Progesterone Receptor Variants Are Associated with Adverse Pregnancy Outcomes

Variants of the gene encoding the progesterone receptor (*PGR)* that are associated with reproductive anomalies have been identified in the human population. One of these variants consists of a 320-base pair Alu element that is inherited in Mendelian fashion, also known as PROGINS [13]. Alu sequences are short and repetitive transposable elements that are primate-specific and comprise up to 11% of the human genome [14]. The PROGINS variant also carries a missense mutation that gives rise to a V660L substitution [15]. There is evidence that the PROGINS variant is less responsive to progestins due to more frequent methylation status and decreased protein stability secondary to phosphorylation and degradation [16]. However, other studies have reported elevated PR transcriptional activity associated with the PROGINS variant, a discrepancy that is likely attributed to experimental design and the use of different cell types [15].

Certain *PGR* polymorphisms are associated with preterm birth; for example, the G allele of *PGR* rs660149 was more frequently detected among Malay women with preterm birth than those with term delivery [17]. Similarly, white and Hispanic women registered in the Utah Population Database that experienced preterm birth were more likely to carry the G allele of *PGR* rs471767 and the GT haplotype across rs471767 and rs578029 than women who delivered at term [18]. *PGR* variants that were identified in fully sequenced Neanderthal genomes have been introduced into the modern human gene pool and are associated with phenotypic traits, such as having more siblings, fewer miscarriages, less bleeding during early pregnancy, or in higher incidence of early pregnancy loss and preterm birth [19,20]. Thus, *PGR* has been subjected to evolutionary changes over time that resulted in consequences to modern-day clinical problems, such as early pregnancy loss and preterm birth.

## 2. Progesterone Receptor Action Controls Decidualization and Is Required for the Successful Establishment and Outcome of a Pregnancy

### 2.1. Alternative Splicing Results in Two PR Isoforms with Distinct Uterine Roles

Progesterone receptor has two isoforms that result from alternative splicing events and give rise to PR-A and PR-B [21]. Unlike PR-A, PR-B contains an additional 165 amino acids in its N-terminus that confers it with a unique transactivation domain. Hence, the presence of its N-terminal transactivation domain allows PR-B to activate a set of target genes that are not shared with PR-A [21,22,23]. This differential function of the PR isoforms can be observed from the divergent phenotypes between the PR-A KO (PRAKO) and PR- B (PRBKO) mouse models. PRAKO mice have decidualization defects that impair embryo implantation along with reduced ovulation [24]. On the other hand, PRBKO mice show a normal uterine and ovarian response to progesterone but develop impaired mammary gland sidebranching and alveogenesis during pregnancy [25]. Despite the absence of PR-A, the transcriptional activity of known PR-B target genes is unchanged in PRAKO mice, indicating a unique functional role for each PR isoform.

Using conditional overexpression mouse models, recent studies have shown that PR-A or PR-B overexpression during the window of implantation perturbs embryo implantation and causes infertility [26,27]. Furthermore, it was also shown that mice with PR-A and PR-B overexpression develop ovarian tumors [28]. There are also isoform-specific roles for PR in the myometrium. For example, PR-B overexpression results in longer gestational length, decreased uterine contractility, and labor dystocia in mice, while PR-A overexpression results in increased uterine contractility with no impact on gestation length [29]. Therefore, extensive evidence from genetic models supports the idea that PR isoforms exert critical roles in the female reproductive tissues of the mouse.

### 2.2. Progesterone Receptor Knockout Mouse Model

Progesterone signaling through PR controls endometrial homeostasis and directs the transcriptional program during decidualization [30]. Studies in mouse models have been critical for driving our understanding of PR function in the uterus and during early pregnancy [31,32]. For example, progesterone receptor knockout (PRKO) mice demonstrated that PR has essential roles in various aspects of reproductive capacity and that female mice are infertile due to a myriad of defects [31]. Specifically, PRKO females were unable to ovulate and had uterine defects such as endometrial hyperplasia, inflammation, and impaired implantation. PRKO mice also showed mammary gland defects, as well as an impaired sexual behavioral response [31].

The inflammatory defects in the PRKO female uterus were possibly arising from the thymic role of PR, as PRKO mice showed abnormal involution during pregnancy [33]. Thymic transplantation models from PRKO to WT mice showed that thymic stromal PR signaling controls fertility by blocking T-cell development during pregnancy, suggesting that progesterone may suppress primary immune cell responses [33,34]. In more recent studies, progesterone was shown to drive the development of thymic regulatory T-cells during pregnancy via activation of the RANK ligand in thymic epithelial cells [35]. Therefore, progesterone action via PR controls key aspects of female reproductive function across various tissues in the body.

### 2.3. The Genomic Landscape of the Progesterone Receptor in Reproductive Tissues

Genome-wide binding studies show that PR controls transcription by binding to the genome as PR-A homodimers, PR-B homodimers, or heterodimers [36]. In the canonical pathway, progesterone induces PR binding at distinct progesterone receptor elements (PRE); however, genome-wide binding studies have revealed that this is not the case and that PR frequently binds at non-PRE regions on DNA [37]. For example, in the mouse uterus, PR binding occurs under basal conditions, and the binding sites increase almost three-fold after acute progesterone treatment [37]. This suggests that while progesterone stimulates most PR binding events, other cellular factors can promote basal PR binding to the DNA. By combining PR binding sites with gene expression microarrays, it was determined that most PR binding sites in the absence of progesterone correlated with repressed gene expression, while PR binding sites in the presence of ligand correlated with both up and down-regulated gene expression. Interestingly, analysis of the distribution of PR binding sites indicated that most binding events localized to distal intergenic and intronic sequences of the genome, while a smaller percentage mapped to upstream promoter regions [37]. It is plausible that the distal PR binding sites control gene expression via complex chromatin looping interactions.

PR binding studies were also performed in T47D breast cancer cells and in primary leiomyoma cells stimulated with the selective progesterone receptor modulator, RU486, which has mixed PR agonist and antagonist activities [38]. More promoter-associated PR binding sites were identified in T47D cells than in the leiomyoma cells (within 5 kb of the transcription start sites), indicating that RU486-mediated binding of PR is highly tissue-specific [38]. Further studies in leiomyoma stem cells show that compared to differentiated leiomyoma cells, *PGR* expression is decreased due to hypermethylation and that this correlated with elevated methylation at PR target genes [39,40]. These results explain the mechanism that underpins leiomyoma tumor regrowth after cessation of anti-progestin therapy and suggest that future therapeutic interventions could be directed to leiomyoma stem cells [41].

### 2.4. PR Binding during Endometrial Stromal Cell Decidualization

Studies in mice have delineated the active role of PR during the peri-implantation period and demonstrated that while PR levels decrease in the luminal epithelium at the time of implantation, they are elevated in the underlying stromal cells [26]. In stromal cells of the mouse endometrium, PR is necessary for inducing and sustaining decidualization [5,31]. Human endometrial stromal cells can be decidualized in vitro by treatment with progestin, medroxyprogesterone acetate (MPA), estradiol, and cyclic AMP (EPC). In decidualizing stromal cells, PR-A and PR-B exert different transcriptional effects, with PR-B regulating a larger portion of the cistrome and transcriptome than PR-A [42,43]. Genome-wide binding analyses of PR reveal that the AP-1 factors, FOSL2 and JUN are regulated by PR and function as transcriptional coregulators during endometrial stromal cell decidualization [44]. PR also interacts with the GATA2 transcription factor during endometrial decidualization, indicating that both pathways are critical during early pregnancy [45].

More recently, studies of PR binding in endometrial biopsies revealed that the transition from the proliferative to the mid-secretory phase of the menstrual cycle corresponds with increased PR binding intervals with unique transcription factor domain preferences, such as FOSL1, FRA1, JUN-AP1, ATF3, and BATF [30]. Transcriptomic profiling of these endometrial tissues indicated enrichment of an inflammatory response during the proliferative to mid-secretory transition. When epithelial-specific PR binding was examined, IRF8 and MEF2C emerged as previously unrecognized factors that are elevated during the peri-implantation phase of the menstrual cycle and may control early pregnancy and embryo implantation [30]. These findings indicate that PR drives a complex transcriptional program that differentially impacts the endometrial stroma and epithelium during the peri-implantation window.

Progesterone receptor is subjected to post-translational modifications, such as sumoylation, that control its activity in decidualizing endometrial stromal cells [46,47]. Upon activation, PR-A is targeted by the small ubiquitin-like modifier (SUMO-1), which modulates its function by altering protein stability and cellular localization. In human endometrial stromal cells, PR-A sumoylation is attenuated during decidualization, a process that stabilizes PR and enhances its function during early pregnancy [48]. Because SUMO-1 can potentially modify several substrates, it is likely that many proteins are post-translationally regulated by this pathway during decidualization, as observed in other reproductive tissues [49,50]. Analysis of the post-translational modifications that occur in decidual cells could uncover the processes that are abnormal during aging, early pregnancy loss, recurrent pregnancy loss, and pre-eclampsia.

### 2.5. Altered PR Function Is Associated with Preeclampsia and Recurrent Pregnancy Loss

Given the crucial role of the progesterone receptor during early pregnancy and decidualization, it is not surprising that pregnancy complications, such as pre-eclampsia, show disrupted PR action [51]. Specifically, a defective decidualization signature in the women who previously developed severe pre-eclampsia identified altered gene networks associated with estrogen receptor (ER) and PR function. Therefore, this altered endometrial PR and ER signature can be used to assess a woman’s risk for developing severe pre-eclampsia.

Recurrent pregnancy loss (RPL) is defined as the loss of three or more consecutive pregnancies before 24 weeks of gestation [52]. RPL is multifactorial, and while it can result from uterine anomalies, chromosomal aberrations, thrombophilia, or immune disorders, over 50% of RPL cases occur due to unknown reasons [53]. Several studies have determined associations between small nucleotide polymorphisms (SNP) in *PGR* among women with recurrent pregnancy loss [54,55]. Specifically, the functional SNP PROGINS (rs1042838) is significantly higher in patients with idiopathic RPL when compared to controls [56]. Another study also showed an association between the +331G/A *PGR* polymorphism and increased failed implantation attempts during in vitro fertilization [57]. These associations suggest that PR action, rather than progesterone levels, may be important mediators of the biology that underpins RPL and can guide future therapeutic interventions.

## 3. Progesterone Withdrawal and the Onset Parturition in Women

### 3.1. Characteristics of Labor Onset in Women

In many mammalian species, parturition is initiated with a “trigger-like” signal composed of several concurrent events, including a drop in serum P4 levels. In humans, however, this trigger model for parturition initiation is less applicable, as serum P4 levels are elevated rather than depleted at the start of labor [58]. Csapo et al. proposed the progesterone withdrawal theory in 1965 to explain how the decrease in P4 signaling that is necessary for uterine contractions could still take place without loss of the steroid hormone’s concentration [59]. The theory suggested that rather than a decrease in progesterone concentration, there was an alternative “block” in its signaling [59]. This, in combination with O’Malley, Sherman, and Toft’s discovery of the Progesterone Receptor in 1970 [60], led to the idea that PR plays an important role in the onset of parturition, and a debate within the scientific community regarding the specifics of this role ensued.

Critical to this debate was O’Malley’s suggestion that PR exists in two isoforms [60], later termed PR-A and PR-B. It is now well-recognized that the relative ratio of PR-A/PR-B is critical for the maintenance of pregnancy and, importantly, that an increase in this ratio within the myometrium is necessary for initiation of parturition [61]. PR-B is thought to promote a relaxed myometrial state, whereas PR-A facilitates contractions through proinflammatory mechanisms [29]. Furthermore, PR-B is the dominant isoform throughout pregnancy, and the subsequent establishment of a positive PR-A/PR-B ratio at parturition is due to a sharp increase in PR-A expression [62]. The exact signaling mechanisms responsible for this switch are a topic of great interest and current investigation due to their high clinical relevance. Because preterm labor is a leading cause of infant mortality and morbidity, improving understanding of how parturition is initiated has significant therapeutic implications [63].

### 3.2. Myometrial PR-B Suppresses Myometrial Contractility

As previously mentioned, PR-A and PR-B share DNA-binding and ligand-binding domains, but PR-A has a shortened N-terminal domain [43,58]. PR-B functions throughout pregnancy to suppress contractility in the myometrium, and it performs this function upon binding to P4. It has been shown in human myometrial cell lines that progesterone-mediated PR-B signaling decreases proinflammatory gene expression when the PR-A/PR-B ratio resembles pregnancy conditions (PR-B dominant). Specifically, PR-B signaling increased the expression of inhibitor-κBα (IκBα), a repressor of the nuclear factor-κB (NFκB) transcription factor, and inhibited basal and lipopolysaccharide-induced proinflammatory gene expression, and these responses were progesterone dependent [64]. Additionally, because PR-B overexpression has been shown to prolong pregnancy, progestin therapies have been intensely investigated for women experiencing preterm labor [29]. At the molecular level, P4 supplementation would bind PR-B and increase its suppressive effects on contractility to prevent preterm labor.

### 3.3. P4 as a Treatment for the Prevention of Preterm Labor

To prolong the suppressive effects of liganded PR-B on contractility, P4 is administered to pregnant women beginning in mid to late gestation. Two forms of progesterone are commonly used for this treatment modality, 17α-hydroxyprogesterone caproate (17-OHPC) or Natural progesterone (P4) [65]. The US Food and Drug Administration (FDA) has approved the use of intramuscular 17-OHCP, but not vaginal P4, to reduce the risk of preterm birth; however, both are commonly used in clinical practice [66]. Qualifications for clinical indication are still a topic of debate; currently, the International Federation of Gynecology and Obstetrics (FIGO) recommendations state that a patient is a candidate for progestogen treatment if they have a prior history of preterm birth (PTB), short cervix (<30 mm), and a singleton pregnancy. In this case, treatment with daily vaginal P4 or weekly intramuscular 17-OHPC treatment is indicated [67].

The initial catalyst for the FDA to approve the use of 17-OHPC was a trial conducted in 2003 by Meis et al. that indicated using this treatment significantly decreased the likelihood of PTB in women with a prior history and a singleton pregnancy [68]. Since its publication, however, numerous critiques of the study’s interpretation and methods have come forth, and subsequent trials—most notably the 2020 PROLONG study have failed to confirm its findings [69]. A notable counterpart to the PROLONG study, the OPPTIMUM study, was a multicenter 2019 trial investigating the efficacy of vaginal P4 treatment in women considered to be at risk for PTB. Here too, P4 was found not to significantly decrease the incidence of premature initiation of parturition [70]. Indeed, in practice, treatment with different forms and dosages of progesterone has had mixed results in preventing preterm labor. The cohort of studies investigating the use of progestogens as a treatment for PTB have been reviewed extensively [65,71,72], yet a consensus has not been reached on under what circumstances this treatment modality is effective or if it is even effective at all.

Among women with singleton pregnancies, there is evidence in support and against that progestogen treatment may significantly impact the incidence of PTB. Across the board, however, it has been found that neither form of progesterone is an effective treatment for reducing the likelihood of PTB in women with multiple pregnancies (e.g., twins) [73,74,75]. This finding is especially concerning because in women with twin pregnancies, the rate of spontaneous early preterm birth is 54.9% as compared to 6.7% for singleton pregnancies [74,76]. Additionally, because history of PTB is a necessary requisite for clinically indicated treatment, women who are pregnant for the first time do not stand to benefit. In summary, 17-OHPC and P4 are not complete nor infallible preventative treatments for women at risk of preterm birth, indicating that our knowledge regarding the molecular basis of P4/PR action during labor onset is not yet complete and requires further investigation.

There are many factors that could be contributing to these suboptimal results, including environmental variables, discrepancies between RCTs, or at the molecular level, the action of PR-A. At the initiation of parturition, the myometrial concentration of PR-A significantly increases while PR-B concentration is maintained, causing the PR-A/PR-B ratio to increase. Importantly, in parturition, PR-A functions in the absence of ligand, starting uterine contractions even in the absence of P4 [58]. Thus, given the unliganded role of PR-A at the onset of parturition, it is not surprising that progesterone treatment is not completely successful in delaying the onset of parturition.

### 3.4. Upregulation of PR-A during Labor

PR-B is the dominant isoform throughout pregnancy until the increase in the PR-A/PR-B ratio in the human myometrium leads to PR-A emergence as the prominent isoform at parturition. Nadeem et al. provided a mechanistic explanation for this switch when they studied the role of PR on Gap-junction alpha protein 1 (*GJAX*/CX43), a critical mediator of uterine contractions [58]. They found that while liganded PR-B was implicated in repressing myometrial CX43, unliganded PR-A functions as a transcriptional activator of the gene encoding CX43 (*GJA1*) [58]. The increase in PR-A expression at parturition is attributed, in part, to a decrease in the myometrial histone deacetylase 1 (HDAC1) expression at parturition. HDAC1 downregulates PR-A expression throughout pregnancy, and HDAC1 decrease during parturition may contribute to the shift in the PR-A/PR-B ratio prior to term or in preterm labor [77].

The transcriptional activation of *CX43* by unliganded PR-A at the onset of parturition was found to occur in both human and mouse myometrium. This is of interest because, at the onset of labor, progesterone (P4) levels are elevated in humans. The lack of P4 binding to PR-A was due to the upregulation of 20-alpha hydroxysteroid dehydrogenase (20αHSD), an enzyme that metabolizes P4 [58]. Thus, efficacy of treatment with exogenous P4 might also be attenuated by this same mechanism, and this model is depicted in Figure 2. A mouse model lacking 20αHSD displayed significantly prolonged gestation periods [78]. However, 20αHSD deficiency partially rescued pregnancy losses in mice with STAT5B deficiency, as STAT5B is a transcriptional repressor of 20αHSD in both mice and humans [78]. Furthermore, increased microRNA-200a expression in the myometrial cells from both humans and mice was correlated with increased metabolism of P4 because miRNA-200a represses STAT5b [79]. A recent study found that there is increased expression of 20αHSD in human myometrium during term labor and in mouse uterus during term and preterm labor. Additionally, the upregulation of 20αHSD, as well as a subsequent decrease in the concentration of active P4, was correlated with proinflammatory stimuli that stimulated labor onset. The authors of this study postulated that, for these reasons, an inhibitor of the 20αHSD protein or its gene *ARK1C1* might be an effective addition to treatment options for PTB [80]. An alternative to inhibiting 20αHSD might be to find a form of progesterone that it cannot metabolize. A selective progesterone receptor modulator that is not metabolized by 20αHSD, promegestone, was recently shown effectively delay term labor and prevent preterm labor in mice because the treatment perpetuated PR signaling [81]. Collectively, these findings suggest that preventative treatment for PTB with progesterone-like molecules that are not metabolizable by 20αHSD warrants further investigation.

## Figures and Tables

**Figure 1 cells-11-01474-f001:**
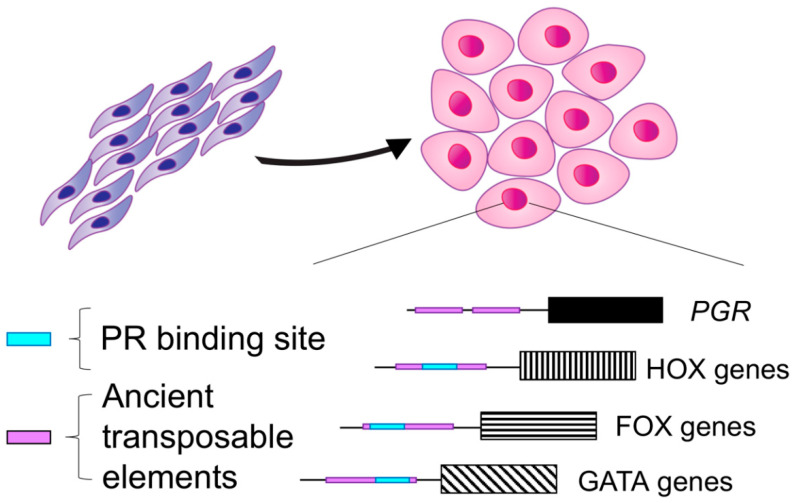
Evolution of progesterone receptor (PR) action in endometrial stromal cells gave rise to decidualization. Decidualization is a feature that evolved in Eutherian mammals and was partly driven by the integration of ancient transposable elements within the gene-regulatory sequences of key masters of decidualization, such as the gene encoding the progesterone receptor (*PGR*) and the HOX, FOX, and GATA gene families. Analysis of PR binding sites in decidual stromal cells indicates the presence of ancient transposable elements within PR binding sites (depicted as the teal rectangles), suggesting that integrations of ancient transposable elements (depicted as purple rectangles) conferred decidualizing stromal cells with a unique sensitivity to P4. The figure depicts endometrial stromal cells in purple and decidual stromal cells in pink.

**Figure 2 cells-11-01474-f002:**
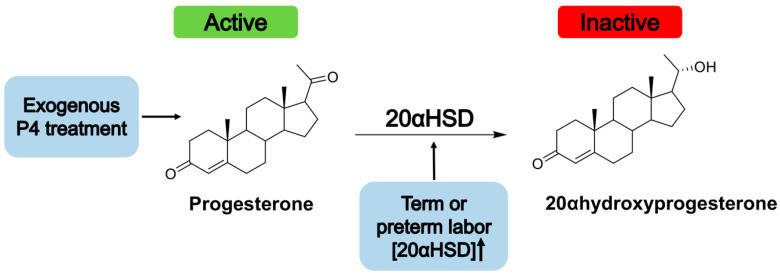
Elevated progesterone metabolism by 20αHSD occurs at the onset of labor. At the onset of parturition, 20αHSD expression is upregulated, and this leads to increased P4 metabolism. This impedes the ability of P4 to bind its ligand, PR-B, and maintain a quiescent pregnancy state. Instead, unliganded PR-A produces proinflammatory effects leading to contractions. For these reasons, exogenous P4 would be largely metabolized and therefore unable to exert the suppressive effects on contractions that the P4-PR-B complex accomplishes.

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
