# Peer review of "Progesterone Receptor Signaling in the Uterus Is Essential for Pregnancy Success"

_cells, 2022, doi:10.3390/cells11091474_

Round 1

Reviewer 1 Report

This is a very well written review by Cope and Monsivais which focuses on progesterone receptor function in decidua and myometrium with emphasis on pregnancy and parturition.  I have but a few minor suggestions.

  1. For the paragraph on PROGINS, could this be focused more so on the PGR polymorphisms associated with uterine tissue/pregnancy complications and not discussing ovarian cancer?
  2. Should the section on the PGR isoforms be moved to an earlier section in the review prior to discussing PGR expression, function?
  3. The section on menstruation (lines 364-388) seems out of place and doesn't fit the overall theme of the review.  Can this be omitted?

Author Response

Response to reviewer’s comments

Reviewer 1

This is a very well written review by Cope and Monsivais which focuses on progesterone receptor function in decidua and myometrium with emphasis on pregnancy and parturition.  I have but a few minor suggestions.

We thank this reviewer for their helpful comments and positive feedback. We have addressed each of the suggestions below and in the text. 

1. For the paragraph on PROGINS, could this be focused more so on the PGR polymorphisms associated with uterine tissue/pregnancy complications and not discussing ovarian cancer?

Thank you for this suggestion. We have removed the section discussing ovarian cancer and focus on the effect of PGR variants in pregnancy.

2. Should the section on the PGR isoforms be moved to an earlier section in the review prior to discussing PGR expression, function?

Thank you, we moved the section related to PGR isoforms so it precedes the discussion about PR function.

3. The section on menstruation (lines 364-388) seems out of place and doesn't fit the overall theme of the review.  Can this be omitted?

We agree with this recommendation and have omitted the section about PR and menstruation.

Reviewer 2 Report

Cope and Monsivais have written a clear and comprehensive review of the role of progesterone receptor signaling on female reproduction via binding to progesterone receptor-A and progesterone receptor-B.  The discussion of the current understanding of the different physiological actions of PR-A and PR-B is first-rate and will be very useful to the reader.  Indeed, this up-to-date review of progesterone action provides important insights into our understanding of progesterone action in the uterus.  Also of importance, is the discussion of open, unresolved questions on progesterone physiology, which may be more important.

Side comment: Due to my interest in evolution, I liked the discussion of the PR variant found in Neanderthals and in some humans. 

In summary: this is a first-rate review.

Although not necessary for this review, which correctly focuses on P4 binding to the PR, I am curious if 11-deoxycorticosterone [21-hydroxyprogesterone] is important in pregnancy.

Minor correction:

Line 249- Toft instead of Tuft.

Author Response

Response to reviewer’s comments

Reviewer 2

Cope and Monsivais have written a clear and comprehensive review of the role of progesterone receptor signaling on female reproduction via binding to progesterone receptor-A and progesterone receptor-B.  The discussion of the current understanding of the different physiological actions of PR-A and PR-B is first-rate and will be very useful to the reader.  Indeed, this up-to-date review of progesterone action provides important insights into our understanding of progesterone action in the uterus.  Also of importance, is the discussion of open, unresolved questions on progesterone physiology, which may be more important. Side comment: Due to my interest in evolution, I liked the discussion of the PR variant found in Neanderthals and in some humans. In summary: this is a first-rate review.

Although not necessary for this review, which correctly focuses on P4 binding to the PR, I am curious if 11-deoxycorticosterone [21-hydroxyprogesterone] is important in pregnancy.

We thank you for your positive comments and interesting question. 11-deoxycorticosterone is synthesized by the action of 21-hydroxylase from progesterone and it is a precursor to aldosterone. Aldosterone is the ligand for mineralocorticoid receptor, which plays a critical role in the regulation of renal salt reabsorption by increasing the activity of the epithelial sodium channel in the nephron. Progesterone is metabolized by 21-hydroxylase and 17-a-hydroxylase into a variety of mineralocorticoid and glucocorticoid molecules. The impact of these metabolites depends on the enzyme isoform abundance, which can vary largely among individuals, and studies have shown associations between elevated glucocorticoid concentrations (PMID 16309788) and 11-deoxycorticosterone (PMID 32274422) and preterm labor. It’s been suggested that the presence of these important progesterone metabolites, and further understanding of their actions on uterine contractility, may help develop novel biomarkers or therapies for pre-term birth.

Minor correction: Line 249- Toft instead of Tuft.

Thank you, this was corrected.